
# Unruh effect for interacting particles with ultracold atoms

Arkadiusz Kosior[1,2], Maciej Lewenstein[2,3] and Alessio Celi[2,4*]

**1** Instytut Fizyki imienia Mariana Smoluchowskiego,
Uniwersytet Jagielloński, Łojasiewicza 11, 30-348 Kraków, Poland
**2** ICFO-Institut de Ciències Fotòniques, The Barcelona Institute of Science
and Technology, 08860 Castelldefels (Barcelona), Spain
**3** ICREA - Pg. Lluís Companys 23, 08010 Barcelona, Spain
**4** Center for Quantum Physics, Faculty of Mathematics, Computer Science and Physics,
University of Innsbruck, and Institute for Quantum Optics and Quantum Information,
Austrian Academy of Sciences, Innsbruck A-6020, Austria

⋆ alessio.celi@gmail.com

## Abstract

The Unruh effect is a quantum relativistic effect where the accelerated observer perceives the vacuum as a thermal state. Here we propose the experimental realization of the Unruh effect for interacting ultracold fermions in optical lattices by a sudden quench resulting in vacuum acceleration with varying interactions strengths in the real temperature background. We observe the inversion of statistics for the low lying excitations in the Wightman function as a result of competition between the spacetime and BCS Bogoliubov transformations. This paper opens up new perspectives for simulators of quantum gravity.

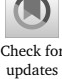
# 1 Introduction

Since the seminal idea of Feynman [1] the capabilities and regimes of applications of quantum simulators [2, 3] have rapidly grown in the last decades. In particular, systems naturally defined on a lattice like trapped ions [4, 5] and neutral groundstate and Rydberg atoms in optical lattices and tweezers [6–8], are excellent platforms for simulating coursed-grained models. They are obviously well suited for simulating condensed matter phenomena where the lattice is intrinsically there. For instance, by engineering synthetic gauge fields by laser means [9] in real or synthetic lattices [10, 11] one can experimentally realize Hofstadter model [12, 13] and visualize the edge states of quantum Hall effect [14–16], as well as realize other topological models like the Haldane model [17]. Similarly by lattice shaking [18], one can achieve non-Abelian gauge fields [19, 20] and observe classical magnetism [21, 22] (for comprehensive reviews see e.g. [23]). Such single-particle simulators are a first step towards experiments that could help solving open questions in many-body phenomena through quantum simulation, as for instance the relation between the Hubbard model and high-$T_c$ superconductivity [24–26]. The combination of synthetic gauge fields with interactions could lead for instance to the realization of (quasi-)fractional quantum Hall states in ladders [27, 28] or quantum spin liquid [29] states in triangular lattices with ultracold atoms.

Lattice based atomic simulators are also well suited for simulating high-energy physics where the lattice is adopted as regularization tool for studying, e.g., strongly coupled gauge theories [30]. Lattice gauge theories can be interpreted as the natural next step after synthetic gauge fields, where the phases representing the classical gauge fields are promoted to operators acting on the gauge degrees of freedom living on the links of the lattice. By adopting convenient gauge invariant truncations of such degrees of freedom, one can design simulators of both Abelian [31–37] and non-Abelian gauge theories [38–40] capable to probe confinement, string breaking, and to study the dynamics of charges as in the first proof-of-principle

experimental realization of the Schwinger model with four ions [41, 42]. Together with the new classical simulation approach based on tensor networks, e.g. [43–50], such simulators offer a new way to study challenging open questions in high-energy physics such as the dynamics and the phase diagram of strong interactions at finite temperature and densities (for review see [51–53]).

Recently, lattice based quantum simulators have been pushed forward also for the study of gravitational phenomena [54]. There is a natural and unresolved tension between quantum physics and general relativity that produces spectacular and counterintuitive effects. Indeed, the quantization requires the choice of a preferred time direction, which fixes the notions of vacuum state and particles, choice that it is at odd with general covariance and makes such notions depending on the observer [55, 56]. Such tension becomes manifest in presence of an event horizon and leads to the appearance of phenomena such as the Hawking radiation of a black hole [57] and the Unruh effect [58] for an accelerated observer. The latter sees the vacuum as a thermal state with a temperature proportional to acceleration essentially because in his/her reference frame (Rindler metric) there is an event horizon and he/she can access only part of the vacuum [59–61]. Simulating the Unruh effect and the Hawking radiation is interesting on one hand because these striking effects are hard to observe in nature, on the other hand because they are among the experimentally accessible phenomena that can bring us closer to quantum gravity.

There are essentially two ways of emulating gravity. The first known as analog gravity exploits the similarity between Navier-Stocks and Einstein-Hilbert equations to engineer the motion of classical and quantum fluids in artificial space-times (see [62–64] for a review). In particular the Unruh effect and the Hawking radiation were studied with phononic quasiparticles in a Bose-Einstein condensate [65–77], photons [78–81] and even classical surface waves on moving water [82, 83] (for very recent ultracold atom experimental analogues of the cosmological expansion see [84, 85]).

Alternatively, one can simulate the motion of artificial Dirac fermions in artificial gravity background by tailoring Dirac Hamiltonian on the lattice properly. This strategy allows for a systematic quantum simulation approach. As shown in [54] (see also [86–88]) one can engineer the motion of Dirac fermions in optical metrics [89] by making the Fermi velocity position dependent, that is by engineering position-dependent tunneling rates on the lattice, which can be equivalently translated in random walk processes [90–93]. One can apply such procedure to any lattice formulation of Dirac Hamiltonian [94], as for instance artificial graphene obtained from ultracold fermions in a brick-wall lattice [95] (see [96, 97] and [98] for realizations of hexagonal optical lattices). Note that one can obtain the motion of Dirac fermions in certain curved space-times by physically bending a graphene sheet [99–102]. In principle, one can observe signatures of the Unruh effect with some subtleties in such a system [103–105] as well as in more complicate scenarios [106].

Engineering artificial gravity through position-dependent tunneling rates for ultracold atoms in optical lattices offers several advantages, as we have shown recently in Ref. [107]:

1. It allows for much more tunability as it is as hard as engineering synthetic gauge fields (position dependent tunneling phases);

2. It allows for a direct experimental observation of the Unruh effect through a quantum quench;

3. It allows to go beyond single-particle phenomena and explore many-body physics in curved space-times.

Here we take a first step in the this largely unexplored world and consider the manifestation of the Unruh effect for interacting fermions in two spatial dimensions. The model we

consider is essentially equivalent to the 2D version of Thirring model [108] (the thermal nature of Hawking radiation was verified for 1D version in [109]). We propose an experimental protocol for studying the interplay between relativistic invariant quartic interactions and the acceleration with ultracold atoms. In the experiment, the acceleration of the (interacting) vacuum can be obtained by quenching the Dirac Hamiltonian as proposed first in [107], while the interactions can be controlled for instance by Feshbach resonance [110]. Alternatively, interaction can be mediated by another bosonic atom, as proposed for the Thirring model in flat space in [111]. We show that when the system can be described thought BCS theory the Unruh effect occurs for the Cooper pairs: due to the "inversion of statics" [60] we observe a crossover between bosonic and fermionic thermal response in two spatial dimensions.

The paper is organized as follows. In Sec. 2 we review the derivation of the single-particle and interacting naive Dirac Hamiltonian in the Rindler metric on a square lattice. The expert reader can directly to Sec. 3 that is the heart of the paper where we discuss the theoretical and experimental aspects of the Unruh effect for interacting Dirac fermions. We consider the case of attractive interactions and derive the Wightman response functions in the mean field limit. We obtain it by determining the Bogoliubov transformation that relates the quasi-particle states in Rindler and Minkowski spacetime, once the normalization of the Rindler Hamiltonian relative to the Minkowski one is chosen such that the former equals the latter in the far horizon limit, i.e. on the boundaries of the cylindrical lattice. We find the expected thermal response function for the quasi-particles. In particular, at low energies the effective description in terms of Cooper pairs translates in a Fermi-Dirac-like Planckian response characteristic of bosons. In the experimental part, we discuss the feasibility of the model and argue that the Wightman response function of the De Witt detector can be, in principle, measured by using one-particle excitation spectroscopy (see also Ref. [107]). In Sec. 4 we take a step back and analyze in details the BCS theory for the naive Dirac Hamiltonian in Minkowski space. In particular, we discuss and explain its subtle dependence on the boundary conditions on finite-size systems (further explanation can be found in the Appedindix that contains also derivations relevant for Sec. 3). Finally, in Sec. 5 we draw conclusions, we resume our results and comment about the new perspectives and developments opened up by our work.

## 2 Derivation of discrete Dirac Hamiltonian

In this section, we present the derivation of the discrete Hamiltonian of interacting massless Dirac fermions in the Rindler Universe in (2+1) dimensions. For the sake of clarity and to be self contained, we first review the derivation of the non-interacting Hamiltonian [54, 112], by starting from the general Lagrangian density for Dirac particles in (d+1) curved spacetimes. We use this section also to fix the notations used in the following sections.

### 2.1 Non-interacting particles

#### 2.1.1 Lagrangian density

The Lagrangian density of a massless non-interacting Dirac spinor $\psi$ in a $(d+1)$ dimensional curved spacetime reads (see, for example [113–115])

$$\mathcal{L}_0 = -i\,\bar{\psi}\gamma^\mu D_\mu \psi, \tag{1}$$

where $\bar{\psi} = \psi^\dagger \gamma^0$ and

$$D_\mu \psi = \partial_\mu \psi + \frac{1}{4} w_\mu^{ab} \gamma_{ab} \psi. \tag{2}$$

Here $D_\mu \psi$ is the covariant derivative of a spinor $\psi$, introduced to guarantee the invariance under the local Poincaré transformations, $w_\mu^{ab}(x)$ is the spin-connection which gauges the Lorentz group, and $\gamma$'s are the gamma matrices in a curved spacetime

$$\{\gamma^\mu, \gamma^\nu\} = 2 g^{\mu\nu}, \quad \gamma^{\mu\nu} = \frac{1}{2}[\gamma^\mu, \gamma^\nu], \quad \gamma_\mu = g_{\mu\nu}\gamma^\nu, \tag{3}$$

where $g^{\mu\nu}$ is the metric tensor $ds^2 = g_{\mu\nu}dx^\mu dx^\nu$. Note that we are using the mostly-positive metric signature, where the norm of timelike vectors is negative. Choosing the opposite signature would require to multiply all gamma matrices by the imaginary unit and, as a consequence, changing the sign of the Lagrangian density.

In $(2+1)$ dimensions $\gamma$'s are $2\times 2$ matrices, and can be expressed in terms of Pauli matrices. A possible choice for the "flat" gamma matrices in Minkowski space is

$$\gamma_0 = i\sigma_z, \quad \gamma_1 = \sigma_y, \quad \gamma_2 = -\sigma_x. \tag{4}$$

Irrespectively of the choice of the gamma matrices, in $(2+1)$ dimensions the product of all $\gamma$'s is proportional to the identity

$$\gamma_0\gamma_1\gamma_2 = -1. \tag{5}$$

### 2.1.2 Spin-connection and vielbein

The relation between gamma matrices in curved spacetimes $\gamma_\mu \equiv \gamma_\mu(x)$ and the flat matrices $\gamma_a$ is given be a vielbein field $e_\mu^a$, i.e. $\gamma_\mu(x) = e_\mu^a(x)\gamma_a$. The vielbein field introduces locally flat Cartesian frame of reference, and it is defined by

$$e_\mu^a(x)\eta_{ab}e_\nu^b(x) = g_{\mu\nu}(x) \quad \text{or} \quad e_a^\mu(x)g_{\mu\nu}e_b^\nu(x) = \eta_{ab}, \tag{6}$$

where $\eta_{ab}$ is the Minkowski metric tensor, $e_a^\mu$ and $e_\mu^a$ are inverse matrices, and $e_a^\mu e_\mu^b = \delta_a^b$. Note that the vielbein $e_\mu^a$ and the spin-connection $w_\mu^{ab}(x)$ fields cannot be independent, as Lorentz translations must relate orthogonal frames in different points. The requirement of a covariantly constant vielbein [116] imposes

$$D_{[\mu}e_{\nu]}^a = \partial_{[\mu}e_{\nu]}^a + w_{[\mu}{}^a{}_b e_{\nu]}^b = 0, \tag{7}$$

where the indices $\mu$ and $\nu$ are antisymmetrized. The spin-connection can be written explicitly using the Christoffel symbols $\Gamma_{\nu\rho}^\mu$

$$w_\mu^{ab} = e_\nu^a \Gamma_{\sigma\mu}^\nu e^{\sigma b} - e^{\nu a}\partial_\mu e_\nu^b, \tag{8}$$

$$\Gamma_{\nu\rho}^\mu = \frac{1}{2}g^{\mu\alpha}\left(\partial_\rho g_{\alpha\nu} + \partial_\nu g_{\alpha\rho} - \partial_\alpha g_{\nu\rho}\right). \tag{9}$$

### 2.1.3 Lagrangian density in the Rindler Universe

We consider $(2+1)$ dimensional static metric of the Rindler Universe [60, 61, 116]

$$ds^2 = -x^2 dt^2 + dx^2 + dy^2, \tag{10}$$

which describes a flat Minkowski spacetime from the point of view of an observed moving with the constant acceleration $a = 1/|x|$.

The Rindler universe is static. Indeed, with the above coordinate choice the Rindler metric (10) is manifestly time-independent, with the time-like Killing vector $B = \partial_t$. The manifest time-translational invariance allows to construct a conserved Hamiltonian function. We can obtain it in a simple manner from the Dirac Lagrangian in the Rindler space.

From (10), we choose the nonzero components of the vielbein as

$$e_t^0 = |x|, \quad e_x^1 = 1, \quad e_y^2 = 1, \tag{11}$$

which produce a particularly transparent form of curved gamma matrices

$$\gamma_t = |x|\gamma_0, \quad \gamma_x = \gamma_1, \quad \gamma_y = \gamma_2. \tag{12}$$

Accordingly with the choice of a vielbien, we can construct the spin connection, which the only nonzero components are

$$w_t^{01} = \frac{x}{|x|} = -w_t^{10}, \tag{13}$$

where the last equality comes from the antisymmetricity of $w_\mu^{ab}$ in its internal (flat) indices. The Dirac Lagrangian density $\mathscr{L}_0$ written explicitly in the (2+1) Rindler Universe becomes

$$\mathscr{L}_0^R = -i\bar{\psi}\left(\gamma^\mu \partial_\mu - \frac{1}{2x}\gamma_0\gamma_2\right)\psi. \tag{14}$$

### 2.1.4 Dirac Hamiltonian

The Hamiltonian density is obtained by the Legendre transformation

$$\mathscr{H} = \frac{\delta\mathscr{L}}{\delta(\partial_t\psi)}\partial_t\psi - \mathscr{L}, \tag{15}$$

which for the Dirac Lagrangian density $\mathscr{L}_0^R$ in the Rindler Universe reads

$$\mathscr{H}_0^R = i\bar{\psi}\left(\gamma^i \partial_i - \frac{1}{2x}\gamma_0\gamma_2\right)\psi. \tag{16}$$

Finally, we obtain the Rindler Hamiltonian by integrating the Hamiltonian density on a space-like hypersurface $H_0^R = \int d\Sigma\,\mathscr{H}_0^R$, where the volume element $d\Sigma = \sqrt{-g}\,dx dy$ includes the metric determinant. It is very convenient to write the Hamiltonian in the fully symmetric form, i.e.,

$$H_0^R = \frac{1}{2}\int d\Sigma\,\mathscr{H}_0^R + \frac{1}{2}\int d\Sigma\left(\mathscr{H}_0^R\right)^\dagger, \tag{17}$$

resulting in

$$H_0^R = \frac{i}{2}\int dx dy\,|x|\left(\sigma_x \partial_x \psi^\dagger + \sigma_y \partial_y \psi^\dagger\right)\psi + \text{H.c.}, \tag{18}$$

where we express the $\gamma$'s in terms of the Pauli matrices (4).

Once written in the form of (18), we can properly discretize the Rindler Hamiltonian simply by replacing integration $\int dx dy$ with summation $d^2\sum_{m,n}$, where $d$ is the lattice spacing, and by replacing the derivatives with finite differences

$$f\,\partial_x h \quad\rightarrow\quad \frac{f_{m+1,n} + f_{m,n}}{2}\frac{h_{m+1,n} - h_{m,n}}{d}, \tag{19}$$

$$f\,\partial_y h \quad\rightarrow\quad \frac{f_{m,n+1} + f_{m,n}}{2}\frac{h_{m,n+1} - h_{m,n}}{d}, \tag{20}$$

which fulfill the Leibnitz rule for differentiation. Finally, the Rindler Hamiltonian on the lattice reads

$$H_0^R = i\sum_{m,n}\left(t_m^x \psi_{m+1,n}^\dagger \sigma_x \psi_{m,n} + t_m^y \psi_{m,n+1}^\dagger \sigma_y \psi_{m,n}\right) + \text{H.c.}, \tag{21}$$

which has the form of a Hubbard Hamiltonian with non-Abelian non-uniform tunneling terms

$$t_m^x = \frac{|m+1|+|m|}{2}, \quad t_m^y = |m|. \tag{22}$$

Note that the Rindler Hamiltonian is scale invariant, as it does not depend on the lattice spacing $d$ ($d^2$ from the volume element cancels due to the replacement $\psi_{m,n} \to d^{-1}\psi_{m,n}$, done to guarantee the normalization condition $\int dx dy \, \psi^\dagger \psi = 1$).

A very similar derivation can be performed to obtain the Dirac Hamiltonian in a flat Minkowski space. In fact, since the Rindler Hamiltonian in the symmetric form (18) does not contain spin connection terms, it suffices to replace the metric determinant with a constant, i.e. $\sqrt{-g} = c$, which consequently leads to a discrete Hubbard Hamiltonian with a constant tunneling $t$ (hereafter we choose $t$ as an energy scale, i.e. we put $t = 1$).

$$H_0^M = i \sum_{m,n} \left( \psi_{m+1,n}^\dagger \sigma_x \psi_{m,n} + \psi_{m,n+1}^\dagger \sigma_y \psi_{m,n} \right) + \text{H.c.}. \tag{23}$$

The Minkowski Hamiltonian (23) should be recovered from the Rindler Hamiltonian (21) in the asymptotic limit of small acceleration $a = \lim_{|m|\to\infty} 1/|m| = 0$. Therefore, we should choose a common energy scale at the boundary of the system. This is done by rescaling the tunneling values (22)

$$t_m^x \to t_m'^x = \frac{|m+1|+|m|}{2M}, \quad t_m^y \to t_m'^y = \frac{|m|}{M}, \tag{24}$$

where $m = -M, \ldots, M$.

## 2.2 Interacting particles

### 2.2.1 Lagrangian density for interacting particles

In order to describe interacting Dirac particles we need to include a nonlinear term in the Lagrangian density (14)

$$\mathcal{L} = \mathcal{L}_0 + \mathcal{L}_{int}. \tag{25}$$

The simplest choices for $\mathcal{L}_{int}$, which would guarantee Lorentz invariance and U(1)-current conservation are

- density-density interaction [117]

$$\mathcal{L}_{int} = -\frac{\lambda}{2} \left( \bar\psi \psi \right)^2, \tag{26}$$

- current-current interaction [108, 118]

$$\mathcal{L}_{int} = \frac{\lambda}{2} \left( \bar\psi \gamma^\mu \psi \right) \left( \bar\psi \gamma_\mu \psi \right). \tag{27}$$

Note that in (3+1) spacetime dimensions, other possible choices would be $\mathcal{L}_{int} = \frac{\lambda}{2} \left( \bar\psi \gamma_5 \psi \right)^2$ or $\mathcal{L}_{int} = \frac{\lambda}{2} \left( \bar\psi \gamma_5 \gamma^\mu \psi \right) \left( \bar\psi \gamma_5 \gamma_\mu \psi \right)$ [119], but in (2+1) dimensions a $\gamma_5$ matrix is trivial - the product of all gamma matrices is proportional to the identity (5), reflecting the well-known fact that chirality is absent in even spatial dimensions.

### 2.2.2 Interacting Dirac Hamiltonian

The Legendre transformation of the full Lagrangian density (25) leads to the Hamiltonian density

$$\mathscr{H} = \mathscr{H}_0 - \mathscr{L}_{int}. \tag{28}$$

Similarly to the non-interacting case, one gets the full Hamiltonian by integration over a space-like hypersurface

$$H = \int d\Sigma \, \mathscr{H} = H_0 + H_{int}, \quad H_{int} = -\int d\Sigma \, \mathscr{L}_{int}. \tag{29}$$

The discretized version of (29) is straightforward, as neither choice of the Lagrangian interaction density (26) - (27) includes derivatives. For the Rindler Universe and density-density interactions (27), we obtain

$$H_{int}^R = \frac{\lambda}{2} \int d\Sigma \left( \bar{\psi} \psi \right)^2 = \sum_{m,n} U_m n_{\uparrow,m,n} n_{\downarrow,m,n} - \sum_{m,n} \frac{U_m}{2} n_{m,n}, \tag{30}$$

where we decompose the lattice spinor operator $\psi_{m,n} = \left[ c_{\uparrow,m,n} \, c_{\downarrow,m,n} \right]$, introduce the number operators $n_{\sigma,m,n} = c_{\sigma,m,n}^\dagger c_{\sigma,m,n}$, $n_{m,n} = \sum_\sigma n_{\sigma,m,n}$, and define $U_m = |m|\lambda/d \equiv |m|U$. Also, we can easily check that the other choice of Lorentz-invariant interactions (27) leads to an equivalent term

$$-\frac{\lambda}{2} \int d\Sigma \left( \bar{\psi} \gamma^\mu \psi \right) \left( \bar{\psi} \gamma_\mu \psi \right) = 3 \sum_{m,n} U_m n_{\uparrow,m,n} n_{\downarrow,m,n} - \sum_{m,n} \frac{U_m}{2} n_{m,n}, \tag{31}$$

as apart from a different coefficient multiplying $n_{\uparrow,m,n} n_{\downarrow,m,n}$, both interaction terms in (2+1) dimensions have the same form. Finally, let us compare (30) with the corresponding interaction Hamiltonian in the Minkowski space

$$H_{int}^M = U \sum_{m,n} n_{\uparrow,m,n} n_{\downarrow,m,n} - \frac{U}{2} \sum_{m,n} n_{m,n}. \tag{32}$$

Imposing that (30) and (32) coincide in the limit of zero acceleration we have to renormalize $U_m$ accordingly to

$$U_m \to U_m' = U_m/M, \tag{33}$$

so that at the lattice boundaries $U_M' = U$ as previously done in the non-interacting case for the tunneling, $t_M'^{x,y} \sim t = 1$.

## 3 Rindler Universe Dirac Fermions with attractive interactions

In this section we consider a model of Dirac fermions with attractive interactions in the mean-field regime. The primary focus of this section is to analyze the power spectrum of the Rindler noise with increasing interaction strength, and show the difference between interacting thermal particles in a flat space and interacting accelerating particles at non-zero Unruh temperature $T_U$. In order to compute it we first find quasi-particle basis in the Minkowski and in the Rindler space for appropriately normalized Hamiltonians defined in the previous section. Indeed, Rindler Hamiltonian has to coincide with the Minkowski one in the limit of zero acceleration. On the lattice, this requirement translates in matching the coupling on the boundaries in $x$ (we exploit the translational invariance along $y$ by taking it periodic such that our lattice is

a cylinder). Then, we find the unitary transformation between the two quasi-particle basis and determine the Wightman function on the Minkowski ground state in the Rindler universe, i.e., as measured after quenching the Hamiltonian from Minkowski to Rindler spacetime. Finally, we study the case where the Minkowski background, in which the observer accelerates, is the thermal state of the interacting Dirac Hamiltonian. We conclude the section by discussing the experimental requirements for observing the Unruh effect in the presence of interactions with ultracold fermions in optical lattices.

## 3.1 Self-consistent Minkowski Hamiltonian

Let us start with writing effective mean-field Hamiltonian in the Minkowski spacetime at physical temperature $T = 0$ and express eigensolutions in terms of quasiparticle modes.

The full Hamiltonian of interacting Dirac particles in the Minkowski spacetime is a sum of noninteracting (23) and interacting (32) terms

$$H^M = H_0^M + H_{int}^M, \tag{34}$$

or explicitly in (2+1) dimensions

$$H^M = i \sum_{m,n} \left( \psi_{m+1,n}^\dagger \sigma_x \psi_{m,n} + \psi_{m,n+1}^\dagger \sigma_y \psi_{m,n} \right) + \text{H.c.} + U \sum_{m,n} n_{\uparrow,m,n} n_{\downarrow,m,n} - \frac{U}{2} \sum_{m,n} n_{m,n}, \tag{35}$$

where $U$ is an attractive interaction strength $U = -|U|$, and $\psi_{m,n} = \begin{bmatrix} c_{\uparrow,m,n} & c_{\downarrow,m,n} \end{bmatrix}$, $n_{\sigma,m,n} = c_{\sigma,m,n}^\dagger c_{\sigma,m,n}$, $n_{m,n} = \sum_\sigma n_{\sigma,m,n}$. We tackle the problem by approximating the interaction term with a mean field averages [120, 121]

$$U n_{\uparrow,m,n} n_{\downarrow,m,n} \approx \left( \Delta c_{\uparrow,m,n}^\dagger c_{\downarrow,m,n}^\dagger + \Lambda c_{\uparrow,m,n}^\dagger c_{\downarrow,m,n} + \text{H.c.} \right) + W n_{m,n}, \tag{36}$$

where we consider all possible terms, i.e.

$$\Delta = -U \langle c_{\uparrow,m,n} c_{\downarrow,m,n} \rangle, \tag{37}$$
$$W = U \langle n_{\sigma,m,n} \rangle, \quad \sigma = \uparrow, \downarrow, \tag{38}$$
$$\Lambda = U \langle c_{\downarrow,m,n}^\dagger c_{\uparrow,m,n} \rangle. \tag{39}$$

Notice that (39) does not conserve spin (or species) of particles and therefore, it is identically zero $\Lambda \equiv 0$. Consequently, the mean field Minkowski Hamiltonian for interacting particles is

$$H_{mf}^M = H_0^M + H_n^M + H_\Delta^M, \tag{40}$$

where $H_0^M$ is a free particle term given by (23), $H_n^M$ is an on-site potential energy term (where we include the chemical potential $\mu$)

$$H_n^M = \sum_{m,n} \left( W - \frac{U}{2} - \mu \right) \psi_{m,n}^\dagger \psi_{m,n}, \tag{41}$$

and $H_\Delta^M$ is a pairing term

$$H_\Delta^M = \Delta \sum_{m,n} c_{\uparrow,m,n}^\dagger c_{\downarrow,m,n}^\dagger + \text{H.c.} = \frac{i\Delta}{2} \sum_{m,n} \psi_{m,n}^\dagger \sigma_y \psi_{m,n}^* + \text{H.c.}. \tag{42}$$

Before we proceed to the eigensolutions of (40), let us stress that $\Delta$ and $W$, given by (37) - (38), are the averages calculated in the ground state $|0\rangle_M$ of $H_{mf}^M$. Consequently, $\Delta$ and $W$ are not external parameters and the eigenvalue problem should be solved self-consistently.

Furthermore, since both Minkowski and Rindler metrics are translation invariant in $y$ direction, we consider the eigenvalue problem on the cylinder (see Appendix A for the analytical solution on a torus in Minkowski space). After performing the Fourier transformation $\psi_{m,n} = 1/\sqrt{N_y} \sum_{k_y} e^{ik_y n} \psi_{m,k_y}$, we write the Hamiltonian in a compact matrix form

$$H_{mf}^M = \frac{1}{2} \sum_{k_y} \Psi_{k_y}^\dagger H_{k_y}^M \Psi_{k_y}, \tag{43}$$

where a spinor $\Psi_{k_y}$ is defined as

$$\Psi_{k_y}^\dagger = \left( \cdots \psi_{m,k_y}^\dagger \ \psi_{m+1,k_y}^\dagger \cdots \psi_{m,\text{-}k_y}^T \ \psi_{m+1,\text{-}k_y}^T \cdots \right), \tag{44}$$

and

$$H_{k_y}^M = \begin{pmatrix} \sigma_x P_x + \epsilon_{k_y} \sigma_y & i\Delta \sigma_y \\ -i\Delta^* \sigma_y & \sigma_x P_x - \epsilon_{k_y} \sigma_y \end{pmatrix}, \tag{45}$$

where $P_x$ is a discrete derivate in $x$ direction $P_x \psi_{m,k_y} = -i\,t\left( \psi_{m+1,k_y} - \psi_{m-1,k_y} \right)$ and $\epsilon_{k_y} = 2\,t \sin k_y$. In the Hamiltonian matrix (45) we drop the on-site potential term (41), as we choose the half-filling condition which yields $W = U/2$ at $\mu = 0$.

Eventually, we can express the spinor $\Psi_{k_y}$ in terms of the quasiparticle eigenmodes, which are eigenvectors of (45)

$$\Psi_{k_y} = \sideset{}{'}\sum_p \left\{ \begin{pmatrix} U_{k_y,p}^M \\ V_{k_y,p}^M \end{pmatrix} \beta_{k_y,p}^M + \begin{pmatrix} V_{\text{-}k_y,p}^{M*} \\ U_{\text{-}k_y,p}^{M*} \end{pmatrix} \beta_{\text{-}k_y,p}^{M\dagger} \right\} = \sideset{}{'}\sum_p \begin{pmatrix} U_{k_y,p}^M & V_{\text{-}k_y,p}^{M*} \\ V_{k_y,p}^M & U_{\text{-}k_y,p}^{M*} \end{pmatrix} \begin{pmatrix} \beta_{k_y,p}^M \\ \beta_{\text{-}k_y,p}^{M\dagger} \end{pmatrix}$$

$$\equiv \sideset{}{'}\sum_p M_{k_y,p} \begin{pmatrix} \beta_{k_y,p}^M \\ \beta_{\text{-}k_y,p}^{M\dagger} \end{pmatrix}, \tag{46}$$

where $\sideset{}{'}\sum_p$ is a summation over positive eigenenergies of the matrix Hamiltonian (45) labeled by $p$, $E_{k_y,p}^M \geq 0$, and

$$X_{k_y,p}^{(+)} = \begin{pmatrix} U_{k_y,p}^M \\ V_{k_y,p}^M \end{pmatrix}, \quad X_{k_y,p}^{(-)} = \begin{pmatrix} V_{\text{-}k_y,p}^{M*} \\ U_{\text{-}k_y,p}^{M*} \end{pmatrix} \tag{47}$$

are column eigenvectors to $E_{k_y,p}^M$ and $-E_{k_y,p}^M$, respectively (see Appendix A). The expression (46) is a canonical Bogoliubov transformation between interacting particles and quasiparticles which diagonalize the Hamiltonian (45).

## 3.2 The mean field Rindler Hamiltonian

In the Sec. 2 we derive a kinetic (21) and interacting (29) terms of the Dirac Hamiltonian in the Rindler Universe $H^R = H_0^R + H_{int}^R$, that explicitly read

$$H^R = i \sum_{m,n} \left( t_m^{'x} \psi_{m+1,n}^\dagger \sigma_x \psi_{m,n} + t_m^{'y} \psi_{m,n+1}^\dagger \sigma_y \psi_{m,n} \right) + \text{H.c.}$$

$$+ \sum_{m,n} U_m' n_{\uparrow,m,n} n_{\downarrow,m,n} - \sum_{m,n} \frac{U_m'}{2} n_{m,n}, \tag{48}$$

where primes denote the rescaled tunnelings (24) and interactions (33). Let us stress again that an accelerated observes moves in the physical vacuum $|\Omega\rangle$, which is a ground state of the Minkowski Hamiltonian $|\Omega\rangle = |0\rangle_M$. Although the Rindler Hamiltonian governs the dynamics

of an accelerated observer, its ground state $|0\rangle_R$ does not obviously need to coincide with the physical vacuum $|0\rangle_R \neq |\Omega\rangle$. Therefore, from the point of an accelerated observed, its background is an excited state.

For that reason, the mean field averages for the interaction terms should now be calculated not in the ground state $|0\rangle_R$, but in the physical vacuum $|\Omega\rangle = |0\rangle_M$. For example, the Rindler pairing function $\Delta_R$ reads

$$\Delta_R(m) = -U'_m \langle c_{\uparrow,m,n} c_{\downarrow,m,n} \rangle = \xi_m \Delta, \tag{49}$$

where $\xi_m = |m|/M$ is a rescaled distance from the horizon.

Apart from the position-dependent elements, the Rindler Hamiltonian (48) has the same form as the Minkowski Hamiltonian (35), and therefore we obtain its mean field counterpart practically automatically

$$H^R_{mf} = \frac{1}{2} \sum_{k_y} \Psi^\dagger_{k_y} H^R_{k_y} \Psi_{k_y}, \tag{50}$$

with

$$H^R_{k_y} = \begin{pmatrix} \sigma_x R(\xi) + \epsilon_{k_y}(\xi) \sigma_y & i\Delta(\xi)\sigma_y \\ -i\Delta^*(\xi)\sigma_y & \sigma_x R(\xi) - \epsilon_{k_y}(\xi) \sigma_y \end{pmatrix}, \tag{51}$$

where $\xi$ is a rescaled position operator $\xi \psi_{m,k_y} = \xi_m \psi_{m,k_y}$, $\Delta(\xi) = \xi \Delta$ is a position-dependent pairing function, $\epsilon_{k_y}(\xi) = \xi \epsilon_{k_y}$ a position-dependent tunneling energy (transverse to the horizon), and $R(\xi)$ is a discrete $|x|P_x$ operator $R(\xi)\psi_{m,k_y} = -it'^x_m \psi_{m+1,k_y} + it'^x_{m-1} \psi_{m-1,k_y}$.

Once solving the eigenvalue equation for $H^R_{k_y}$, one can express the spinor $\Psi_{k_y}$ (44) in terms of the Rindler quasiparticles

$$\Psi_{k_y} = \sum_p{}' \begin{pmatrix} U^R_{k_y,p} & V^{R*}_{-k_y,p} \\ V^R_{k_y,p} & U^{R*}_{-k_y,p} \end{pmatrix} \begin{pmatrix} \beta^R_{k_y,p} \\ \beta^{R\dagger}_{-k_y,p} \end{pmatrix} \equiv \sum_p{}' R_{k_y,p} \begin{pmatrix} \beta^R_{k_y,p} \\ \beta^{R\dagger}_{-k_y,p} \end{pmatrix}, \tag{52}$$

where we remind the reader that the prime in summation indicates that $p$ in $\sum_p'$ runs over the positive eigenvalue, $E^R_{k_y,p} > 0$.

Now, let us focus on how the Unruh temperature $T_U$ influences the Rindler pairing $\Delta_R$ (in Sec. 4 we discuss how the pairing $\Delta$ in the Minkowski space changes with the physical temperature $T$). Since $\Delta_R(m) = \xi_m \Delta$ and $\xi_m$ is proportional to the inverse of acceleration $1/a = |m|$, we could be tempted to write that the $\Delta_R$ is proportional to the inverse of Unruh temperature $T_U = a/(2\pi)$. Nevertheless such conclusion would not be correct. The Unruh temperature is defined locally on $x = constant$-hypersurfaces of the Rindler Universe and therefore is different for inequivalent observers. The expression (49) tells us how the pairing function changes with $|m|$ from the point of view of the Minkowski observer. Since the proper time slows down the closer we are to the horizon, then the interactions seem to be weaker. Therefore $\Delta_R(m)$ for the observer at $|m|$ should be rescaled with the proper time $\Delta_R(m)/\xi_m = \Delta$. Consequently, the pairing strength seen by an accelerated observer does not depend on the Unruh temperature. At the same time, an observer at $|m|$ sees the pairing to be weaker $\Delta_R(m)/\xi_{m'} < \Delta$ when she/he looks towards the horizon and stronger $\Delta_R(m)/\xi_{m'} > \Delta$ when she/he looks opposite to the horizon.

### 3.3 Wightman function for an accelerated observer

The expressions (46) and (52) are canonical Bogoliubov transformations [120, 121] between interacting fermionic particles and noninteracting quasiparticles in the Minkowski and Rindler

spacetimes, respectively. As quasiparticles modes (46) and (52) diagonalize the mean-field Hamiltonian (43) and (50), we can write

$$
H_{mf}^A = \frac{1}{2} \sum_{k_y} \Psi_{k_y}^\dagger H_{k_y}^A \Psi_{k_y} = \frac{1}{2} \sum_{k_y} {\sum_p}' E_{k_y,p}^A \left( \beta_{k_y,p}^{A\dagger} \beta_{k_y,p}^A - \beta_{-k_y,p}^A \beta_{-k_y,p}^{A\dagger} \right) =
$$
$$
\sum_{k_y} {\sum_p}' E_{k_y,p}^A \beta_{k_y,p}^{A\dagger} \beta_{k_y,p}^A + const., \tag{53}
$$

where $p$ in the sum runs over positive eigenvalues and $A$ refers to either Minkowski ($M$) or Rindler ($R$). Note that in the last equality of (53) we use the fact that $H_{\pm k_y}^A$ have the same spectra (see the discussion of the symmetries of $H_{k_y}^A$ in Appendix C).

From (53) we see that the groundstate of $H_{mf}^A$ is deprived of quasiparticle excitations. Therefore, it is anihilated by all quasiparticle operators

$$
\beta_{k_y,p}^A |0_A\rangle = 0, \quad \forall k_y, p \text{ s.t. } E_{k_y,p}^A > 0. \tag{54}
$$

A state that fulfills (54) is of a form

$$
|0_A\rangle \propto \prod_{k_y,p,E_{k_y,p}^A<0} \beta_{k_y,p}^{A\dagger} |0\rangle, \tag{55}
$$

where $|0\rangle$ where is a particle vacuum. Since quasiparticle operators mix particle creation and annihilation processes, we directly see that the ground state $|0_A\rangle$ must contain particles, and that the groundstates of Minkowski and Rindler Hamiltonians are different as they have different quasiparticle excitations. Also, because quasiparticle modes (46) and (52) are different for the two (stationary and accelerated) observers, the act of creation of a particle is seen differently. Combining (46) and (52) together and using $R_{k_y,p}^\dagger R_{k_y',p'} = \delta_{k_y,k_y'} \delta_{p,p'}$ we obtain

$$
\begin{pmatrix} \beta_{k_y,p}^R \\ \beta_{-k_y,p}^{R\dagger} \end{pmatrix} = {\sum_{p'}}' R_{k_y,p}^\dagger M_{k_y,p'} \begin{pmatrix} \beta_{k_y,p'}^M \\ \beta_{-k_y,p'}^{M\dagger} \end{pmatrix}, \tag{56}
$$

which is the spacetime Bogoliubov transformation [60,61] between two noninteracting quasiparticles from the point of view of different observers. It is straightforward to find out that in general $\beta_{k_y,p}^R |0_M\rangle \neq 0$, and so indeed $|0_M\rangle \neq |0_R\rangle$. In particular, we expect a different response from a particle detector for different observers. Let us write the Wightman function for an accelerated observer

$$
G_{m,n}(t) = \langle 0_M | c_{\sigma,m,n}^\dagger(t) c_{\sigma,m,n} |0_M\rangle, \tag{57}
$$

and its Fourier time transform

$$
G_{m,n}(\omega) = \int \mathrm{d}t\, e^{-i\omega t} G_{m,n}(t), \tag{58}
$$

which is the power spectrum of the Rindler noise.

Because the time evolution $c_{\sigma,m,n}^\dagger(t)$ is different for the stationary and accelerated observers, and since $|0_M\rangle \neq |0_R\rangle$, it is intuitive that a response function $G_{m,n}(\omega)$ should also be different for the two reference frames. However, the most interesting part is far less intuitive: (i) the response function $G_{m,n}(\omega)$ for an accelerated observed exhibits thermal behavior, (ii) in even spacetime dimensions thermal distribution of fermions (bosons) is Fermi-Dirac (Bose-Einstein), but in odd spacetime dimensions the statistic interchange. (Let us stress that the latter does not imply a violation of the canonical anticommutation/commutation relations,

but it is an apparent statistic interchange that comes from dimensional differences in wave propagation known as the Takagi inversion theorem [60]. In Ref. [107] we verified the interchange of statistics for noninteracting fermions with a dimensional crossover.) In particular, in the continuous limit the power spectrum of a thermal noninteracting gas in (2+1) Minkowski spacetime is [60]

$$G_M^T(\omega) = \begin{cases} |\omega| |e^{\omega/T} + 1|^{-1} & \text{(fermions)} \\ |e^{\omega/T} - 1|^{-1} & \text{(bosons)} \end{cases}, \tag{59}$$

whereas an accelerated observer sees

$$G(\omega) = \begin{cases} |\omega| |e^{\omega/T_U} - 1|^{-1} & \text{(fermions)} \\ |e^{\omega/T_U} + 1|^{-1} & \text{(bosons)} \end{cases}, \tag{60}$$

where the Unruh temperature is $T_U = a/(2\pi)$.

One might wonder how to relate the fermionic and bosonic power spectra of a cold thermal gas ($T \to 0$) in a flat space, to the ones seen by an accelerated observer in the limit of zero acceleration ($T_U \to 0$). In fact, it is easy to find out that in the zero temperature limit, the modulus of the Bose distribution is Fermi-Dirac

$$\lim_{T \to 0} |e^{\omega/T} - 1|^{-1} = \begin{cases} 0 & , \omega > 0 \\ 1 & , \omega < 0 \end{cases}, \tag{61}$$

therefore, except for a singular point at $\omega = 0$, the power spectra match exactly in the zero temperature limit.

Applying the spacetime Bogoliubov transformation (56) we get explicitly the Wightman function

$$G_m(t) \equiv G_{m,n}(t) = \frac{1}{N_y} \sum_{k_y} \sideset{}{'}\sum_{p,p'} \left( u_{\sigma,m,k_y,p}^{R*}(t) \Gamma_{k_y,p,p'}^{(1)} + v_{\sigma,m,-k_y,p}^{R}(t) \Gamma_{k_y,p,p'}^{(2)} \right) v_{\sigma,m,-k_y,p'}^{M*}, \tag{62}$$

where $u_{\sigma,m,k_y,p}^{R/M}$ and $v_{\sigma,m,k_y,p}^{R/M}$ are the elements of column vectors $U_{k_y,p}^{R/M}$ and $V_{k_y,p}^{R/M}$, respectively,

$$
\begin{aligned}
u_{\sigma,m,k_y,p}^{R}(t) &= e^{-iE_{k_y,p}^R t} u_{\sigma,m,k_y,p}^{R}, \\
v_{\sigma,m,k_y,p}^{R}(t) &= e^{-iE_{k_y,p}^R t} v_{\sigma,m,k_y,p}^{R},
\end{aligned}
\tag{63}
$$

and

$$
\begin{aligned}
\Gamma_{k_y,p,p'}^{(1)} &= U_{k_y,p}^{RT} V_{-k_y,p'}^{M} + V_{k_y,p}^{RT} U_{-k_y,p'}^{M}, \\
\Gamma_{k_y,p,p'}^{(2)} &= U_{-k_y,p}^{R\dagger} U_{-k_y,p'}^{M} + V_{-k_y,p}^{R\dagger} V_{-k_y,p'}^{M}.
\end{aligned}
\tag{64}
$$

Since we expand the Dirac fields in the eigenmodes of the Rindler Hamiltonian, we expect that the Fourier transform of (62) should express the power spectrum as seen by the accelerated observer with $a = 1/|m|$. However, this statement is true only if $t$ is the proper time of the observer. In the Rindler Universe the proper time of an observer is $\xi_m t$, and consequently, to compare the response of different observers, the power spectrum of noninteracting particles needs to be rescaled as $\xi_m G_m(\omega/\xi_m)$ [107]. Furthermore, in the interacting problem we have a band gap separating valance and conductance bands. The band gap $2\Delta_R(m)$ seen by a Minkowski observer is $|m|$ dependent. To account for that, we need to compare shifted power spectra, i.e. $\xi_m G_m\big((\omega - \Delta_R(m))/\xi_m\big)$.

Note that in the continuous limit, the Unruh temperature of noninteracting fermions is $T_U = 1/(2\pi|m|)$. The numerical results for the finite lattice system show that indeed $T_U$

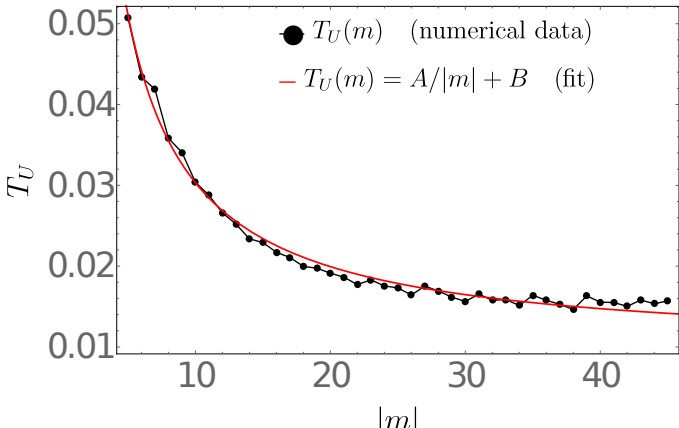

Figure 1: (color online) The Unruh temperature $T_U$ in a finite lattice system for the observer moving with a constant acceleration $|a| = 1/|m|$. We consider here a square lattice on a cylinder (open boundary conditions along $x$) of lengths $100 \times 100$, with the Rindler horizon laying on the circle along $y$ at the half of the cylinder. The numerical values $T_U(m)$ where found from fitting a continuum limit curve (60) to the noninteracting numerical power spectrum for $\omega/\xi < 1$. To the numerical results we fit $T_U(m) = A/|m| + B$ and find $A \approx 0.2$ and $B \approx 0.01$. Thus, the behavior of $T_U(m)$ in our finite lattice is remarkably close to the continuum result, $T_U(m) = 1/(2\pi|m|)$, predicted by the Bisognano-Wichmann theorem.

decreases with increasing $|m|$, but the functional dependence $T_U(m)$ might be in principle different. We investigate the relation $T_U(m)$ by fitting the continuum limit power spectrum (59) to the numerical results. Eventually, we find that $T_U(m) = A/|m| + B$ with $A \approx 0.2$ and $B \approx 0.01$ reproduces quite well the data for $5 < |m| < 45$, see Fig. 1. Thus, we find good agreement with the results of Bisognano-Wichmann theorem [122,123] (see also [124]) that holds in the infinite lattice limit. Indeed, the theorem, which follows from axiomatic field theory, implies that all Lorentz-invariant local field theories display the Unruh effect with $T_U = 1/(2\pi|m|)$. In the following we discuss in details the behavior of the Rindler noise in the different regimes of interest, extending the results obtained by us in [107] in the non-interacting Dirac fermions. As in [107] the numerical results are in the agreement with the continuum limit in the linear dispersion regime, i.e. $\omega/\xi < 1$.

### 3.4 Power spectrum at $T_U \approx 0$ and $T = 0$

We start by discussing the limit of zero acceleration, $T_U \to 0$, in which we expect the Wightman function in the Rindler space to coincide with the Wightman function in the Minkowski space at $T = 0$. By using the analytical solution for the Minkowski system on a torus (see Appendix A), we find for the latter

$$G_M^{T=0}(t) = \langle 0_M | c_{\sigma,m,n}^\dagger(t) c_{\sigma,m,n} | 0_M \rangle = \frac{1}{2} \sum_{\vec{k}} e^{-iE_{\vec{k}}t}, \tag{65}$$

and consequently

$$G_M^{T=0}(\omega) = \frac{1}{2} \sum_{\vec{k}} \delta(\omega + E_{\vec{k}}) \propto \rho(\omega), \tag{66}$$

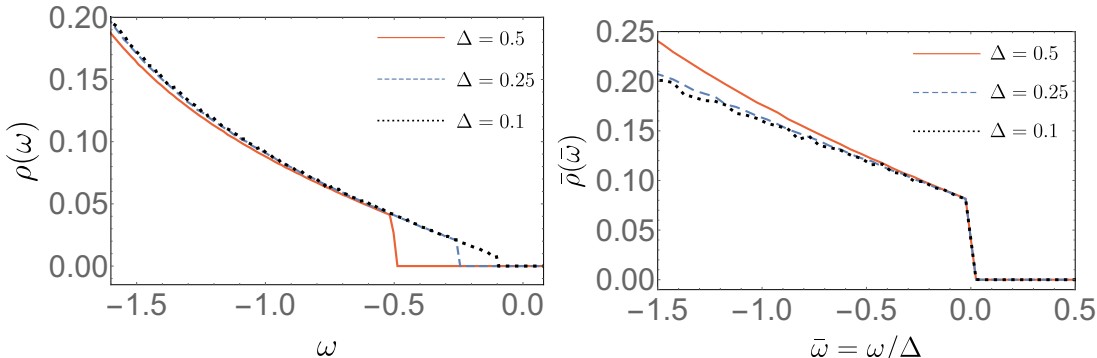

Figure 2: (color online) The numerical density of states $\rho(\omega)$ (left panel) and the rescaled densities $\bar{\rho}(\bar{\omega})$ (right panel) for a $\Delta$-paired Minkowski system on a torus with a quasiparticle dispersion relation (67). We consider here a square lattice on a torus (periodic boundary conditions along both $x$ and $y$) of lengths $5000 \times 5000$. For $|\omega| > \Delta$ the density of states is basically free fermionic. The rescaled density of states curves $\bar{\rho}(\bar{\omega}) = \rho(\omega - \Delta)/\Delta$ overlap for $\bar{\omega} \approx 0$.

where

$$E_{\vec{k}} = \sqrt{\Delta^2 + 4\left(\sin^2(k_x) + \sin^2(k_y)\right)}, \tag{67}$$

is a positive quasiparticle eigenenergy of a system and $\rho(\omega)$ is the negative energy density of states.

The numerical density of states $\rho(\omega)$ for different values of $\Delta$ is plotted on Fig. 2 (left panel). For the noninteracting system the density of states can be well approximated with the continuous limit density, i.e. $\rho(\omega) \propto |\omega|$ for $|\omega| \lesssim 1$. For the interacting system we can approximate

$$G_M^{T=0}(\omega) \propto \rho(\omega) = \mathcal{N}^{-1}|\omega|\,\theta(-\omega - \Delta), \tag{68}$$

for $|\omega| \lesssim 1$, i.e. a nonzero value of the pairing function $\Delta$ introduces a $2\Delta$ band gap and a $\Delta$ step jump in the density of states. As $\Delta$ is much smaller than the half band width $w = 2\sqrt{2} \approx 2.83$ of the noninteracting system, for $|\omega| \gtrsim \Delta$ the density of states is basically free fermionic, and only at $|\omega| \approx \Delta$ encounters the step function jump. As we expect that $G_M^{T=0}(\omega)$ replicates the power spectrum of the Rindler noise for $T_U = 0$, we can interpret this Fermi-Dirac like behavior as a response of composite bosons (i.e. Cooper pairs), which should be fermionic (statistic inversion in even spatial dimensions). In other words the pairing influences power spectrum only near $|\omega| \approx \Delta$, which is expected as Cooper pairs tend to form near the Dirac cones (corresponding to $E_{\vec{k}} \approx \Delta$) and their number increases with $\Delta$ (see Appendix B).

Since we expect that (68) estimates the power spectrum of the Rindler noise for $T_U = 0$ can infer that for $T_U \gtrsim 0$ the qualitative behavior of the power spectrum should be

$$G(\omega) \propto \frac{|\omega|}{e^{(\omega + \Delta)/T_U} + 1}, \tag{69}$$

for $|\omega| \lesssim 1$. It is trivial to realize that (69) in the limit $T_U \to 0$ recovers (68). Also, for $\Delta \approx 0$ and $\omega/T_U \gg 1$ we can drop the plus one in denominator (69) and therefore recover the power spectrum of noninteracting fermions (60).

In order to compare the power spectra for different values of $\Delta$, we need to compensate for different densities of states of the interacting Minkowski ground states (a physical vacuum).

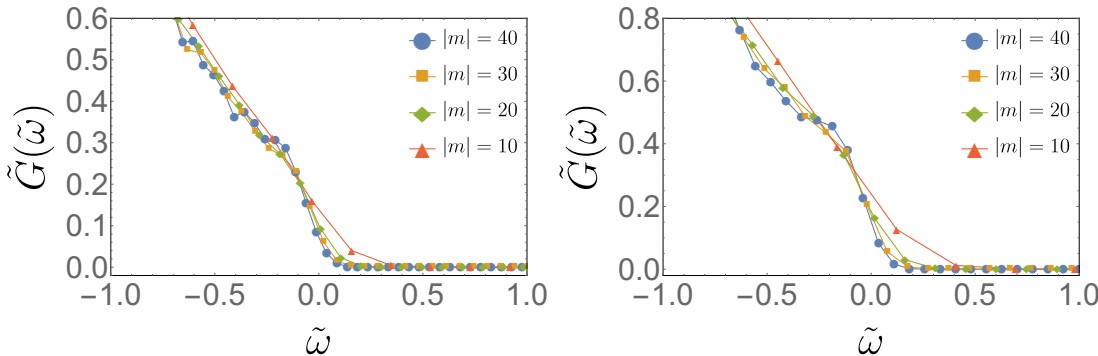

Figure 3: (color online) The power spectra of the Rindler noise, i.e. the Fourier transform of the Wightman function for the interacting Dirac fermions at T=0 and the pairing function $\Delta \approx 0.25$ (left), $\Delta \approx 0.5$ (right). We consider here a square lattice on a cylinder (open boundary conditions along $x$) of lengths $100 \times 100$, with the Rindler horizon laying on the circle along $y$ at the half of the cylinder. For each value of the $|m|$, the $\tilde{G}(\tilde{\omega})$ are convoluted with a Gaussian filter of 2-site width centered around $m$. For $\Delta \approx 0.25$ (left) the power spectrum is close to one of noninteracting fermions (60), but with an indication of a Fermi-Dirac plateau. The $|m| = 40$ curve (red) corresponds to the smallest Unruh temperature and therefore we observe an almost step-like sharp response at low energies. Also, the plateau is more evident for $\Delta \approx 0.5$ (right), since the number of Cooper pairs becomes significant, see Appendix B. The Fermi-Dirac profile is destroyed close to the horizon. Since the density of Cooper pairs is the greatest near the Dirac points, they dominate lowest energy excitations (close to $\tilde{\omega} = 0$). For $|\tilde{\omega}| \gg 0$ the fermionic power spectrum is recovered. Note that the curves were rescaled to account for different proper times of different observers and shifted by $\Delta$ (due to the spectral gap).

After rescaling $\omega$ and shifting the argument of $\rho(\omega)$

$$(\omega, \rho(\omega)) \rightarrow (\bar{\omega}, \bar{\rho}(\bar{\omega})) = (\omega/\Delta, \rho(\omega - \Delta)/\Delta), \tag{70}$$

we find that all $\bar{\rho}(\bar{\omega})$ curves overlap near $\bar{\omega} \approx 0$, see Fig. 2 (right panel). Consequently, (69) after rescaling (70)

$$\bar{G}(\bar{\omega}) = \Delta^{-1} G(\omega - \Delta) \propto \frac{\bar{\omega}}{e^{\bar{\omega}/(T_U/\Delta)} + 1}, \tag{71}$$

for $\bar{\omega} \approx 0$. Note that $\bar{G}(\bar{\omega})$ depends only on the ratio $\alpha = T_U/\Delta \propto 1/(\Delta|m|)$, which might be interpreted as the effective Unruh temperature of the interacting accelerated gas. As a result, as long as $\alpha$ is constant, two nonequivalent accelerated observers might observe the same spectrum $\bar{G}(\bar{\omega})$ near $\bar{\omega} \approx 0$. This behavior is expected, since the interaction term of Rindler Hamiltonian (30) is invariant under the rescaling: $U \rightarrow U c$ and $|m| \rightarrow |m|/c$. Consequently, in the regime when interaction dominates over kinetic energy the power spectra $\bar{G}(\bar{\omega})$ with constant $|m|\Delta$ should coincide.

## 3.5 Power spectrum at $T_U > 0$ and $T = 0$

In this section we calculate the power spectra of the Rindler noise using the explicit formula (62) for the Wightman function. As discussed in Sec. 3.3, we rescale the power spectra according to the proper time of an observer $\tilde{G}(\tilde{\omega}) = \xi_m G_m(\tilde{\omega} - \Delta)$, where $\tilde{\omega} = \omega/\xi_m$. The numerical results are presented on Fig. 3, where we plot the spectra $\tilde{G}(\tilde{\omega})$ for different pairing

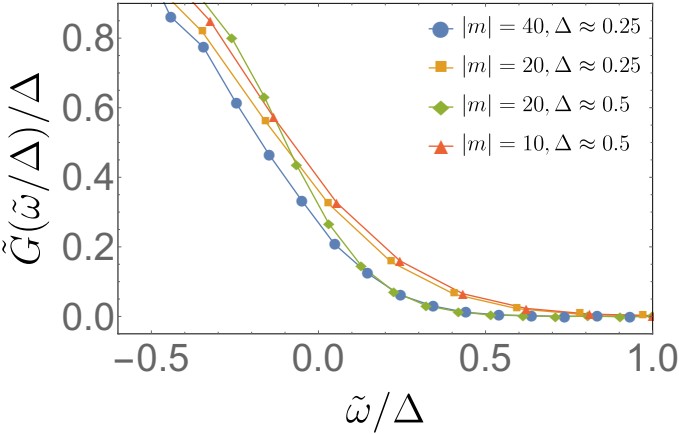

Figure 4: (color online) The power spectra rescaled to account for both proper-time difference of inequivalent observers and the difference in the density of states (70). We consider here a square lattice on a cylinder (open boundary conditions along $x$) of lengths $100 \times 100$, with the Rindler horizon laying on the circle along $y$ at the half of the cylinder. For each value of the $|m|$, the $\tilde{G}(\tilde{\omega})$ are convoluted with a Gaussian filter of 2-site width centered around $m$. Because of the invariance of the interaction term in Rindler Hamiltonian (30) under the rescaling, $U \to U c$ and $|m| \to |m|/c$ (see the discussion in the main text), we expect that when pairing dominates over kinetic energy, the power spectra with $|m|\Delta = $ constant, should be identical. Indeed, blue and green curves ($|m|\Delta = 10$) , as well as red and orange curves ( $|m|\Delta = 5$) overlap close to $\tilde{\omega}/\Delta = 0$.

strengths $\Delta \approx 0.25$ (left panel), $\Delta \approx 0.5$ (right panel) and various distances to the horizon $|m| = 10, 20, 30, 40$. In order to both minimize lattice artifacts and to mimic realistic measurement schemes of finite space resolution, as in [107] we present the $\tilde{G}(\tilde{\omega})$ convoluted with a Gaussian filter of 2-site width.

The numerical results reproduce quite well the expected thermal behavior (69) with the Unruh temperature inversely proportional to the distance to the horizon $T_U \propto 1/|m|$. For $|\tilde{\omega}| \gg 0$ the power spectra are approximately linear (just like in a free fermionic case), while near $|\tilde{\omega}| \approx 0$ we observe a clear Fermi-Dirac profile (i.e. a bosonic response), which is more evident for $\Delta \approx 0.5$, since the number of Cooper pairs is more significant. Note that, as expected, the Fermi Dirac plateau is nearly twice higher for $\Delta \approx 0.5$ then in $\Delta \approx 0.25$.

After rescaling (71) we can directly compare different power-spectrum curves. The results are plotted on Fig. 4. It turns out, that the postulated scaling is indeed valid near $|\tilde{\omega}| \approx 0$, see the discussion in the previous section.

Note that the chosen values of the pairing function $\Delta$ on Fig. 3 are one order of magnitude greater than the order of the Unruh temperature. For $\Delta \sim T_U$ we find that the power spectra are changed only marginally in comparison to the noninteracting ones.

## 3.6 Power spectrum at $T_U > 0$ and $T > 0$

In this section we again consider the power spectrum of Dirac fermions in the Rindler Universe, although, now we consider as background the thermal state at $T > 0$ of the interacting Dirac Hamiltonian in Minkowski space. Such situation is relevant also from the experimental point of view as the ultracold fermions in optical lattices have typically a non-negligible temperature compared to the band width, thus order one in our units ($t = 1$). It is therefore

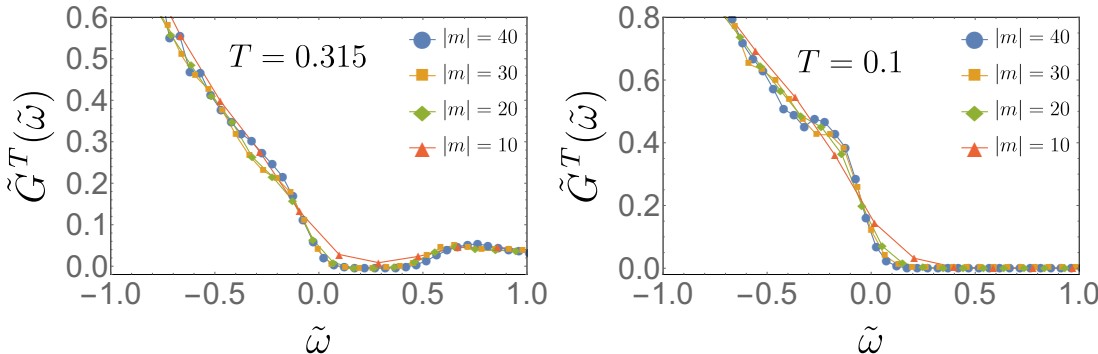

Figure 5: (color online) The power spectra of interacting Rindler Dirac fermions for a thermal background (74) at $T = 0.315$ (left panel) and $T = 0.1$ (right panel). We consider here a square lattice on a cylinder (open boundary conditions along $x$) of lengths $100 \times 100$, with the Rindler horizon laying on the circle along $y$ at the half of the cylinder. For each value of the $|m|$, the $\tilde{G}(\tilde{\omega})$ are convoluted with a Gaussian filter of 2-site width centered around $m$. The temperature $T$ values were chosen such that $\Delta(T = 0.315) \approx 0.25$ and $\Delta(T = 0.1) \approx 0.5$ in order to directly compare with the results in Fig. 3 at $T = 0$. As expected, the effect of a thermal background becomes more significant when the ratio $T/\Delta(T)$ becomes of order one.

crucial to establish the interplay between the Unruh temperature $T_U$ and the physical temperature $T$ such to determine the visibility of the Unruh effect for interacting Dirac fermions, as done in [107] for the noninteracting case. (Also, in the Appendix D we argue that deviations from zero chemical potential have smaller impact on the Wightman function then the physical temperature $T$.) The thermal Wightman function can be written as

$$G_m^T(t) = \text{Tr}[\rho_M(T)c_{m,n}^\dagger(t)c_{m,n}], \tag{72}$$

where

$$\rho_M(T) = \sum_{k_y,p} n(E_{k_y,p}^M)\beta_{k_y,p}^{M\dagger}|0_M\rangle\langle 0_M|\beta_{k_y,p}^M \tag{73}$$

and the power spectrum is

$$G_m^T(\omega) = \int \mathrm{d}t e^{-i\omega t} G_m^T(t). \tag{74}$$

In order to compare with the results of the previous section, we choose the interaction $U = 3.55$ and manipulate the physical temperature $T$ in such a way to obtain $\Delta(T) \approx 0.5, 0.25$ (as in Fig. 3).

The numerical results are presented on Fig. 5. As expected, we find that the thermal background does not affect the Unruh effect until $\Delta(T)/T \sim 1$. In particular, for $T = 0.315$ and $\Delta(T = 0.315) \approx 0.25$ (left panel) we observe a thermal positive frequency contribution from the quasiparticles above the Fermi sea. For $T = 0.1$ and $\Delta(T = 0.1) \approx 0.5$ (right panel) the power spectrum is very close to the $T = 0$ case.

## 3.7 Experimental realization and detection of Unruh effect

In [107] we presented a detailed proposal how to simulate the noninteracting Dirac Hamiltonian in Minkowski (35) and Rindler (48) spacetimes in the optical lattice setup. Since the tunneling matrices of a two component spinor $\psi_{m,n}$ is purely off-diagonal, then flipping its

components at every second site $\psi_{m,n} \rightarrow \sigma_x \psi_{m,n}$ diagonalize the tunneling and the two components do not mix. Therefore, the noninteracting naive Dirac Hamiltonian on a square lattice is equivalent to two independent copies of a $\pi$-flux Hamiltonian, and the optical lattice simulation can be done with one component only. In other words, one can exploit that the lattice is bipartite and that the unit cell has dimension 2 due to the $\pi$ flux to encode the two components of Dirac spinor in two different sublattices (and reduce the doubling of Dirac points to 2). The key feature of our experimental proposal in [107] is that the tunneling is assisted in both $x$ and $y$ directions by Raman lasers that induce a synthetic magnetic field of flux $\pi$ in the symmetric gauge. The intensity of the tunneling is controlled by the intensity of (one of) Raman lasers. Such a scheme allows both for the preparation of the Minkowski vacuum as groundstate of the corresponding non-interacting Dirac Hamiltonian with uniform tunneling rates, and for its "acceleration" via a sudden quench of the tunneling rates to a $V$-shape profile of the Dirac Hamiltonian in Rindler spacetime.

The generalization of the scheme above to the case of interacting Dirac fermions requires the introduction of spatially tunable interactions. One possibility would be to realize the scheme in [107] with dipolar fermionic gases like erbium, where it has been very recently experimentally demonstrated that dipolar interactions are stable and tunable by Feshbach resonances [125]. In such a scheme, each of the two components of the spinor $c_\uparrow$, $c_\downarrow$ are identified with the two spin species of erbium that occupy two different sublattices. The interaction term $n_\uparrow n_\downarrow$, thus becomes a nearest-neighbor density-density interaction provided by the magnetic dipole moment of the atoms. The magnetic field (or in alternative the light shift) inducing the Feshbach resonance has to be then tuned spatially on the lattice spacing scale such to provide the desired $V$-shape interaction profile. In alternative, at the price of doubling the number of Dirac points, we can consider two spin states of fermionic atoms with Feshbach resonance like potassium [126] loaded in a spin-independent $\pi$-flux square lattice and perceiving the same laser-induced tunneling term. The interactions are now on site. A third possibility would be to incorporate in the set-up in [107] an additional lattice that hosts bosonic atoms that mediate the interaction between fermions in the same spin state as in [111]. Again the interaction between the bosonic and fermionic species needs to possess a Feshbach resonance that allows to tune the scattering length appropriately.

In [107] we proposed an experimental scheme to measure $G(\omega)$ by using the one-particle excitation spectroscopy [127], which corresponds to a frequency-resolved transfer of the atoms to initially unoccupied auxiliary band of negligible width. In the weak-coupling limit the number of atoms transferred to the auxiliary band as a function of frequency detuning $\omega$ is proportional to $G(\omega)$. A similar scheme can be adopted here if the interactions in the auxiliary excited state are negligible.

# 4 Pairing function $\Delta(U, T)$

In Sec. 3.2 we argue that the pairing strength $\tilde{\Delta}_R = \xi_m^{-1} \Delta_R$ seen by an accelerated observer does not depend on the Unruh temperature, and is the same for all inequivalent observers in the Rindler Universe $\tilde{\Delta}_R = \Delta$. In this section we shall consider how the Minkowski pairing function $\Delta$ depends on the physical temperature $T$ and the interaction strength $|U|$.

At the half filling for the noninteracting system $U = 0$, we have $\Delta = 0$ and the ground state of the Minkowski Dirac Hamiltonian (23) is a Dirac semimetal with two valence and conduction bands touching at Dirac points [128–132]. Similarly to a standard BCS theory [120, 121], in a half-filled attractive Fermi Hubbard model on a square lattice even arbitrarily small attractive interactions $U < 0$ give rise to a nonzero $\Delta$ pairing [133–135]. On the contrary, it is known that below a critical interaction strength $|U_c| \neq 0$, Dirac fermions on a honeycomb

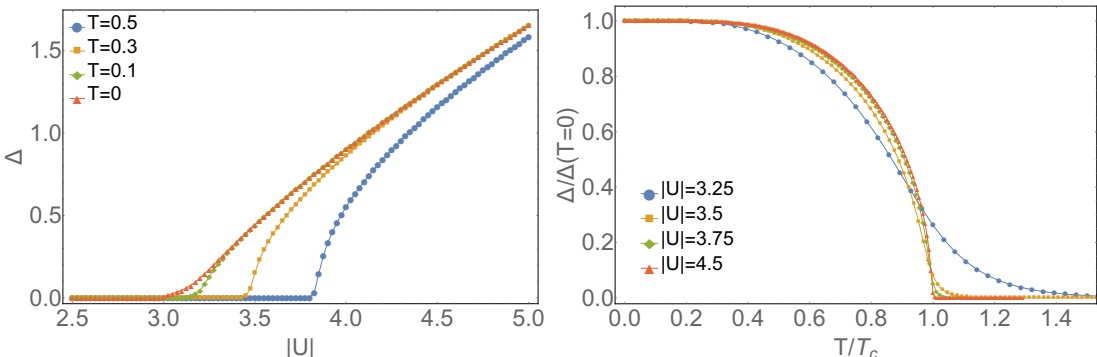

Figure 6: (color online) The meanfield pairing function $\Delta$ for a Dirac Hamiltonian (35) on a cylinder as a function of the interaction strength $|U|$ (left panel), and the temperature $T$ (right panel). We consider here a square lattice on a cylinder (open boundary conditions along $x$) of lengths $100 \times 100$. In particular, we find that: (i) $U_c$ increases with the temperature $T$ (left), (ii) the temperature phase transition is more evident when the interaction strength $|U|$ increases (right). Near $T = T_c$ for $U \gg U_c$ qualitatively recover the crltial behavior known from the standard BCS theory, see the main text.

lattice do not form Cooper pairs, since the density of states vanishes linearly at Dirac points [136–140]. At $T = 0$ and at the critical interaction $|U_c|$ the system undergoes the quantum phase transition between semimetal and a paired superconductor [141], although different methods, i.e. mean-field, variational and Monte Carlo give different estimates of the critical interaction $|U_c| \sim 2 - 5$. Note that honeycomb and square lattices are bipartite and therefore the particle-hole transformation allows us to relate attractive and repulsive Fermi Hubbard models at the half-filling [142].

Here, we analyze $\Delta(U, T)$ dependence for two types of boundary conditions: (i) open in $x$ and periodic in $y$ (cylinder), (ii) periodic in both $x$ and $y$ (torus). We find that at $T = 0$ the boundary conditions have a strong affect the pairing properties of finite systems. In particular, we show that arbitrarily small $U < 0$ gives rise to pairing in a small finite system on a torus.

## 4.1  Solution on a cylinder

Here we study in details the behavior of BCS pairing function for the Hamiltonian (35) on a cylindrical square lattice. For concreteness, we present and discuss the numerical results for a cylinder of size $50 \times 50$ lattice system (open in $x$ direction).

In Fig. 6 (left panel) we plot the pairing gap as a function of the interaction strength for several values of a physical temperature $T$. Our numerical results show that the pairing properties of the system described by the Hamiltonian (35) are similar to the Hubbard model on the honeycomb lattice, as one could expect. The quantitative differences might be due to the different dispersion relations away from the Dirac points. At $T = 0$ we recover the critical value $|U_c| \approx 3$ as obtained for an attractive $\pi$-flux model [131]. For $T > 0$, the critical interaction increases. In Fig. 6 (right panel), we plot the temperature dependence of the pairing function. As expected, we find that the pairing gap $\Delta(T)$ diminishes with the increasing temperature, and at some point we reach a normal unpaired state. In finite systems, it is always the crossover. Nevertheless, we observe that with increasing $U$ the behavior of $\Delta(T)$ starts to resemble a phase transition known from the standard BCS theory. It is known that in the standard BCS

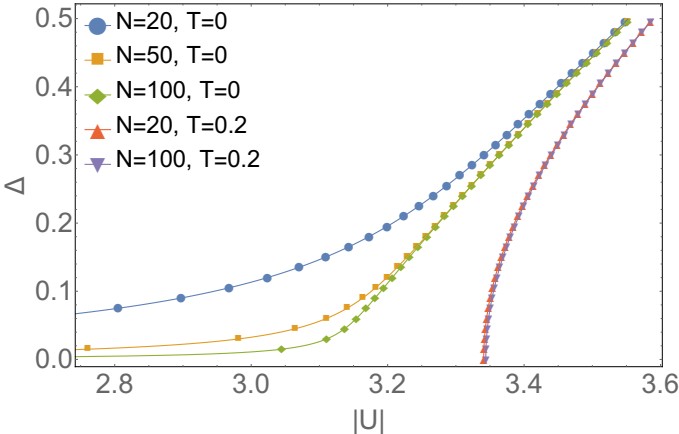

Figure 7: (color online) The meanfield pairing function $\Delta$ for a Dirac Hamiltonian (35) on a torus as a function of the interaction strength $|U|$ for different system sizes $N$ and temperatures $T$. We consider here a square lattice on a torus of lengths $N \times N$. The numerical results are in agreement with (79). For $T \approx 0$ and $N = 20$ the critical interaction is zero, but already for $N = 100$ the pairing $\Delta(U)$ approaches the thermodynamic limit. For $T = 0.2$ the finite size correction in (79) is negligible, the two curves for $N = 20$ and $N = 100$ overlap.

theory, the near critical behavior is universal [143]

$$\Delta(T \approx T_c) \approx A\sqrt{1 - T/T_c}, \tag{75}$$

and

$$\Delta(T \approx 0) \approx B T_c, \tag{76}$$

where $A \approx 3.07\, T_c$, $B \approx 1.764\, T_c$, $A/B \approx 1.74$ are independent of material. We find that the numerical results on Fig. 6 (right panel) qualitatively reproduce the standard BCS theory, with (75) being a good approximation to the critical transition region. Quantitatively, the parameters $A$ and $B$ tend to the BCS values with increasing interaction strength $|U|$. In particular, we find $A \approx 2.40\, T_c$, $B \approx 1.49\, T_c$, $A/B \approx 1.62$ for $|U| = 3.75$, and $A \approx 2.93\, T_c$, $B \approx 1.68\, T_c$, $A/B \approx 1.75$ for $|U| = 4.5$. One may question the validity of the meanfield approach for values of the interactions of the order of the band width. While we can expect (small) quantitative deviations with respect to more precise approaches like diagrammatic quantum MonteCarlo, the qualitative behavior is known to be well captured by the mean field approach.

## 4.2 Solution on a torus

Here we analyze a system described by the Hamiltonian (35) in the meanfield approximation on a torus. Interestingly, contrary to the previous result, in this case we find analytically that for finite systems, the pairing does happen for any $|U|$ at $T = 0$. We comment more on this point at the end of this section.

In this case the expression for $\Delta$ can be computed analytically (see the Appendix A for details) and reads

$$\Delta = \frac{|U|}{N_x N_y} \sum_{\vec{k}} \frac{\Delta}{2 E_{\vec{k}}} \tanh\left(\frac{E_{\vec{k}}}{2T}\right), \tag{77}$$

where

$$E_{\vec{k}} = \sqrt{\Delta^2 + 4\left(\sin^2(k_x) + \sin^2(k_y)\right)}, \tag{78}$$

is a positive quasiparticle eigenenergy of a system. From (77) we get

$$|U_c(T)| = \lim_{\Delta \to 0+} \left( f(\Delta, T) + \frac{2}{\Delta N_x N_y} \tanh\left(\frac{\Delta}{2T}\right) \right)^{-1} = \left( f(0, T) + \frac{1}{N_x N_y T} \right)^{-1}, \qquad (79)$$

where

$$f(\Delta, T) = \frac{1}{N_x N_y} \sum_{\vec{k} \neq D.P.} \frac{1}{2E_{\vec{k}}} \tanh\left(\frac{E_{\vec{k}}}{2T}\right) \qquad (80)$$

is finite in the thermodynamic limit $N_x, N_y \to \infty$. The summation in (80) goes over all $\vec{k}$ in the Brillouin zone except for $\vec{k} \in \{(0\ 0), (0\ \pi), (\pi\ 0), (\pi\ \pi)\}$, since for $\Delta = 0$ the dispersion relation has four Dirac points which require a separate treatment as $E_{\vec{k}} = 0$.

First of all, let us analyze the equation (79) in thermodynamic limit. Taking $N_x, N_y \to \infty$ we get

$$|U_c(T)| \overset{N \to \infty}{=\!=\!=} f(0, T)^{-1}, \qquad (81)$$

which is finite for all $T < \infty$ and reproduces the numerical results for a system on a cylinder. Indeed, for $T \approx 0$ we obtain

$$|U_c(T \approx 0)| \overset{N \to \infty}{=\!=\!=} f(0, 0)^{-1} \approx 3.1. \qquad (82)$$

Instead, for finite systems we have

$$|U_c(T)| \overset{N < \infty}{=\!=\!=} \left( f(0, T) + \frac{1}{N_x N_y T} \right)^{-1}, \qquad (83)$$

where now the second term cannot be dropped. In particular, $|U_c(T)| \propto T$ for small $T$ and goes to zero in the limit $T \to 0+$. (Note that the order of limits is important: if one puts $T = 0$ in (79) before taking the limit $\Delta \to 0+$ then the finite size result would be different.)

Because $\lim_{T \to 0+} |U_c(T)| = 0$, the pairing of Dirac fermions on torus can take place for an arbitrarily small interaction strength $|U|$. This apparent discrepancy between the two finite-size solutions of the Dirac Hamiltonian (23) with different boundary conditions can be explained as following. From (77) at $T = 0$ the highest contribution to $\Delta$ comes from the smallest positive eigenvalues $E_{\vec{k}}$. In particular, for small $\Delta \approx 0$ the highest contribution comes from the Dirac cones $\vec{K} \in \{(0\ 0), (0\ \pi), (\pi\ 0), (\pi\ \pi)\}$ (the density distribution of Cooper pairs is discussed in details in the Appendix B) which greatly affects small systems but is irrelevant in the thermodynamic limit, see Fig. 7. At the same time, this argument does not apply to the solution on the cylinder, as for even number of points $N$ the two bands touch only in a thermodynamic limit, *cf.* Fig. 6.

Different than for the naive Dirac Hamiltonian on a square lattice (23), the boundary conditions do not play a significant role for the honeycomb lattice, where the critical interactions is nonzero $|U_c| > 0$ also on a torus. Again, such behavior admits a simple explanation in terms of the band structure of the model. The graphene dispersion relation

$$E_{\vec{k}} = \sqrt{3 + 2\cos(k_y \sqrt{3}) + 4\cos(\sqrt{3}/2k_y)\cos(3/2k_x) + \Delta^2}, \qquad (84)$$

is minimalized only in the thermodynamic limit as the Dirac cones' coordinates $\vec{K} = \pm(0, 4\pi/(3\sqrt{3}))$ are not integer multiple of $\pi$. Thus, as it happens for finite cylindrical lattices the Dirac points do not contribute to the computation of $\Delta$ also in finite toroidal honeycomb lattices.

# 5 Conclusions and Outlook

In this paper we have investigated and proposed the experimental observation of the Unruh effect for interacting particles with ultracold fermions in optical lattices by a quantum quench. We have shown that achieving tunable Lorentz-preserving interactions with fermionic atoms is possible. Thus, it is possible to simulate an accelerated observer in an interacting background by simulating the corresponding Hamiltonian in Rindler space. Observing the Unruh effect reduces then to measuring the Wightman response function in Rindler spacetime for the interacting background, the ground state of the interacting Dirac Hamiltonian in Minkowski space. While here and in [107] we consider the detection of the Wightman function by one particle excitation spectroscopy as witness of the thermal behavior, one can in principle search for signatures of Unruh effect in other correlation functions, for instance density-density correlations. This is an interesting research direction we plan to pursuit in the next future.

We have studied the Wightman response function detected by an accelerated observer for attractive relativistic interactions in the meanfield approximation, for varying interactions and real temperature $T$ of the background. In this approximation, the Unruh effect results from the interplay between the two different Bogoliubov transformations that relate the notion of particles for inertial and accelerated observers, and of particles and BCS quasi-particles, respectively. In the low-energy limit, in which the lattice system is with good approximation relativistic invariant, we have found that the Wightman function (precisely its power spectrum) displays the Planckian spectrum characteristic of the Unruh effect with a peculiarity. When the interactions grow, up to dominate over the Unruh temperature, there is a crossover between normal and "double" inversion of statistics determined by the (bosonic) Cooper pairs. Remarkably, in the low-energy limit our meanfield lattice calculations for interacting fermions not only give that the response is thermal, but also that the functional relation between the Unruh temperature $T_U$ and the proper acceleration $a$ is the same as for a free theory, $T_U = 1/(2\pi a)$.

Such finding is in agreement with the predictions of the Bisognano-Wichmann theorem, which are valid under very general assumptions for any relativistic quantum field theory. On one hand, such fast convergence of lattice calculations to the correct relativistic behavior indicates that our lattice approach to the quantum simulation of quantum field theories in curved spacetime is promising. On the other hand, it can be read as a further evidence of the robustness of the Bisognano-Wichmann predictions recently observed in [144], where the equivalence between entanglement Hamiltonian and the Hamiltonian perceived by accelerated observers is used to access the entanglement spectrum of lattice models.

The present paper opens up interesting perspectives. It offers for instance a natural setup for testing quantum thermometry [145] and quantum thermodynamics [146] in curved spacetimes in presence of interactions (for a review on recent trends and developments in quantum thermodynamics see e.g. [147]).

It offers also a experimental playground for toy models of Lorentz-violating and trans-Planckian physics [148] (see also [62] for a comprehensive review and references therein). Indeed, as we argue above there is a tight relation between Unruh effect and basic principles of relativistic quantum field theory as Lorentz invariance and locality. This tight relation explains the universality of the thermal behavior of the Wightman function [149, 150]. Conversely, deviations from such behavior can signal e.g. the breaking of Lorentz invariance [151], the deformation of the uncertainty principle [152], or even open a window on quantum gravity [153]. Ultracold atom simulators of quantum interacting matter in artificial curved spacetime may serve for analyzing/testing such scenarios (*cf.* with Lorentz violations in neutrino physics [154, 155]).

Beyond the Unruh effect, another interesting direction for the quantum simulator we propose here is the study of dynamical chiral symmetry breaking in curved spacetime [156]. In-

deed, by considering further atomic species we can in principle include more than one flavour and engineer the Gross-Neveu model [157] (see also a recent coldatom quantum simulator of the (1+1)d lattice Gross-Neveu model [158]) in an arbitrary optical (or more complicated) metric. Last but not least, our study can be seen a further important step in the long journey to the simulation of self-gravitating quantum manybody physics.

## Acknowledgements

We thank Leticia Tarruell and Javier Rodríguez-Laguna for fruitful discussions.

**Funding information**  A.K. acknowledges a support of the National Science Centre, Poland via Projects No. 2016/21/B/ST2/01086 and 2015/16/T/ST2/00504. M.L. acknowledges the Spanish Ministry MINECO (National Plan 15 Grant: FISICATEAMO No. FIS2016-79508-P, SEVERO OCHOA No. SEV-2015-0522), Fundació Cellex, Generalitat de Catalunya (AGAUR Grant No. 2017 SGR 1341 and CERCA/Program), ERC AdG OSYRIS, EU FETPRO QUIC, and the National Science Centre, Poland-Symfonia Grant No. 2016/20/W/ST4/00314. A.C. acknowledges financial support from the ERC Synergy Grant UQUAM and the SFB FoQuS (FWF Project No. F4016-N23)

## A  Analytic equation for $\Delta$ on torus

In this section we seek for a fully periodic solution of the mean-field Minkowski Hamiltonian at the half filling. After expressing the field operators in the momentum space, we can write

$$H_{mf}^M = \frac{1}{2} \sum_{\vec{k}} \begin{pmatrix} \psi_{\vec{k}}^\dagger & \psi_{-\vec{k}}^T \end{pmatrix} H_{\vec{k}}^M \begin{pmatrix} \psi_{\vec{k}} \\ \psi_{-\vec{k}}^* \end{pmatrix}, \tag{85}$$

where $\psi_{\vec{k}}^\dagger = \begin{pmatrix} c_{\uparrow,\vec{k}}^\dagger & c_{\downarrow,\vec{k}}^\dagger \end{pmatrix}$, and

$$H_{\vec{k}}^M = \begin{pmatrix} \vec{g}_{\vec{k}} \circ \vec{\sigma} & i\Delta\sigma_y \\ -i\Delta\sigma_y & \vec{g}_{\vec{k}} \circ \vec{\sigma}^* \end{pmatrix}, \tag{86}$$

where $\Delta = |U|\langle c_{\uparrow,m,n} c_{\downarrow,m,n}\rangle$, $\vec{g}_{\vec{k}} = 2t\left(\sin(k_x)\ \sin(k_y)\right)$, and $\vec{\sigma} = \left(\sigma_x\ \sigma_y\right)$, such that $\vec{g}_{\vec{k}} \circ \vec{\sigma} = 2t\left(\sin(k_x)\sigma_x + \sin(k_y)\sigma_y\right)$. The eigenvalues of (86) read

$$\lambda_{\vec{k}} = \pm\sqrt{\Delta^2 + 4\left(\sin^2(k_x) + \sin^2(k_y)\right)} \equiv \pm E_{\vec{k}}. \tag{87}$$

The two eigenvectors associated to the positive eigenvalue are

$$X_{\vec{k},1}^{(+)} = \frac{1}{\sqrt{2}} \begin{pmatrix} 1 \\ G_{\vec{k}} \\ 0 \\ \frac{\Delta}{E_{\vec{k}}}, \end{pmatrix}, \quad X_{\vec{k},2}^{(+)} = \frac{1}{\sqrt{2}} \begin{pmatrix} 0 \\ -\frac{\Delta}{E_{\vec{k}}} \\ 1 \\ G_{\vec{k}}^* \end{pmatrix}, \tag{88}$$

with $G_{\vec{k}} = \frac{2t}{E_{\vec{k}}}\left(\sin(k_x) + i\sin(k_y)\right)$. Let us the write the field operator explicitly in terms of the eigenmodes

$$\begin{pmatrix} \psi_{\vec{k}} \\ \psi_{-\vec{k}}^* \end{pmatrix} = \sum_{p=1,2}\left(X_{\vec{k},p}^{(+)}\beta_{\vec{k},p} + X_{\vec{k},p}^{(-)}\beta_{-\vec{k},p}^\dagger\right), \tag{89}$$

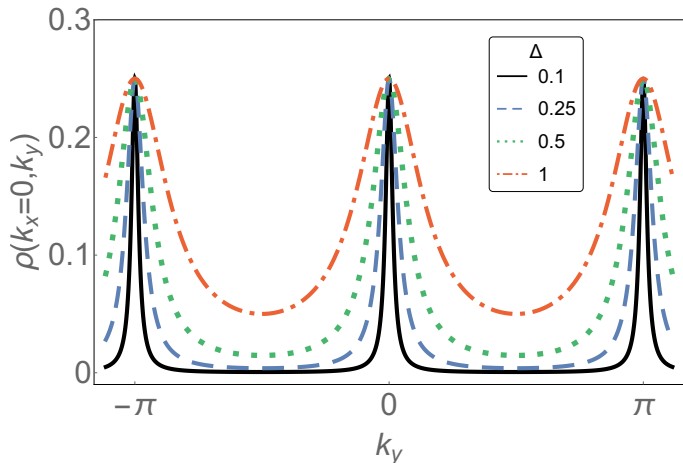

Figure 8: (color online) The plot of Cooper pairs density $\rho(k_x = 0, k_y)$ obtained from (91). Up to $\Delta \approx 1$, with good approximation the Cooper pairs are formed around Dirac cones only.

where $X_{\vec{k},p}^{(-)} = (\sigma_x \otimes \mathbb{1}_2) X_{-\vec{k},p}^{(+)*}$.

Finally, by using the decomposition (89) we can obtain an equation for the pairing function

$$\Delta = \frac{|U|}{N_x N_y} \sum_{\vec{k}} \langle c_{\uparrow,\vec{k}} c_{\downarrow,-\vec{k}} \rangle = \frac{|U|}{N_x N_y} \sum_{\vec{k}} \frac{\Delta}{2 E_{\vec{k}}} \tanh\left( \frac{E_{\vec{k}}}{2\,T} \right). \tag{90}$$

## B  Pairs density

From (90), the quantity $\rho(\vec{k}) = |\langle c_{\uparrow,\vec{k}} c_{\downarrow,-\vec{k}} \rangle|^2$ can be interpreted as the density of Cooper pairs in the whole system. For $T = 0$

$$\rho(\vec{k}) = |\langle c_{\uparrow,\vec{k}} c_{\downarrow,-\vec{k}} \rangle|^2 = \frac{\Delta^2}{4 E_{\vec{k}}^2}, \tag{91}$$

where

$$E_{\vec{k}} = \sqrt{\Delta^2 + 4\left(\sin^2(k_x) + \sin^2(k_y)\right)}. \tag{92}$$

It is straightforward to conclude that for $\Delta \approx 0$ the density of Cooper pairs is very small $\rho(\vec{k}) \approx 0$, unless $E_{\vec{k}} \approx \Delta$. This is the case for the Dirac points $\vec{K} \in \{(0\,0), (0\,\pi), (\pi\,0), (\pi\,\pi)\}$, where the density is maximal, i.e., $\rho(\vec{K}) = 1/4$. This simple observation tells us that the pairing is dominated by Dirac-like excitations near the Dirac cones. This is the reason why the bosonic response (the Fermi Dirac plateau) only appears in the low frequency regime (Fig. 3). We plot $\rho(k_x = 0, k_y)$ in Fig. 8. The figure shows that up to $\Delta \approx 1$, with good approximation the composite bosons (Cooper pairs) are formed around Dirac cones only.

## C  Symmetries of $H_{k_y}^{M/R}$

In this part we consider all possible symmetries of the matrix Hamiltonian $H_{k_y}^A$, where $A$ stands both for $R$ (Rindler) or $M$ (Minkowski). First of all, it is easy to find out that a unitary operator

$$U_1 = \sigma_x \otimes \mathbb{1}_X \otimes \mathbb{1}_2, \tag{93}$$

where $\mathbb{1}_X$ is the $N_x \times N_x$-identity matrix over the lattice coordinate $m$ along $x$, transforms the Hamiltonian as

$$U_1^\dagger H_{k_y}^A U_1 = -H_{-k_y}^{A*}. \tag{94}$$

Thanks to this symmetry, in (46) and (52) we are able to obtain the negative energy eigenmodes out of positive solutions $X_{\vec{k},p}^{(-)} = U_1 X_{-\vec{k},p}^{(+)*}$. Additionally, after making a gauge choice such that $\Delta \in \mathbb{R}$, then we see that there is another symmetry of the Hamiltonian

$$U_2^\dagger H_{k_y}^A U_2 = -H_{-k_y}^A, \tag{95}$$

where

$$U_2 = \sigma_y \otimes \mathbb{1}_X \otimes \sigma_z. \tag{96}$$

At the half filling, we also have

$$U_3^\dagger H_{k_y}^A U_3 = -H_{k_y}^{A*}, \tag{97}$$

where

$$U_3 = \sigma_z \otimes \mathbb{1}_X \otimes \mathbb{1}_2. \tag{98}$$

All four possible combinations of $U_1$, $U_2$, $U_3$, i.e. $U_1 U_2$, $U_1 U_3$, $U_2 U_3$ and $U_1 U_2 U_3$ are also necessarily symmetries of $H_{k_y}^A$. In particular, defining $U_4 = U_1 U_3$ we see

$$U_4^\dagger H_{k_y}^A U_4 = H_{-k_y}^A, \tag{99}$$

therefore, we conclude $H_{k_y}^A$ and $H_{-k_y}^A$ have the same spectrum.

# D   Non-zero chemical potential

In this section we show that a moderate non-zero chemical potential is not critical for the observation of the Unruh effect in optical lattices. A non-zero chemical means that what we are preparing is not the vacuum but a state that it is slightly populated by particles or antiparticles, depending on the sign of the chemical potential.

We expect that the Planckian signature in the Wightman response function to be robust against chemical potentials up to values of the order of the Unruh temperature. Such situation is indeed not so different than starting with a gas of atoms at non-zero temperature. As we show for both interacting (see Sec.3.6) and non-interacting [107] Dirac fermions, distinctive features of the Planckian spectrum survive up to temperature of the order or even slightly higher than the Unruh temperature itself. We also expect that the finite temperature of the sample will be in general larger in typical ultracold atom experiments than the error in the calibration of the atom filling, or that at worst they are of the same order. Thus, we conclude that the essential requirement for the experiment is that they are at most of the order of the maximal Unruh temperature, which is a fraction of the band width (see Fig.1).

We back the arguments raised above with the numerical study of a non-zero chemical potential for non-interacting fermions (as argued above we expect a similar behavior also in presence of interactions). The numerical results are plotted in Fig. 9.

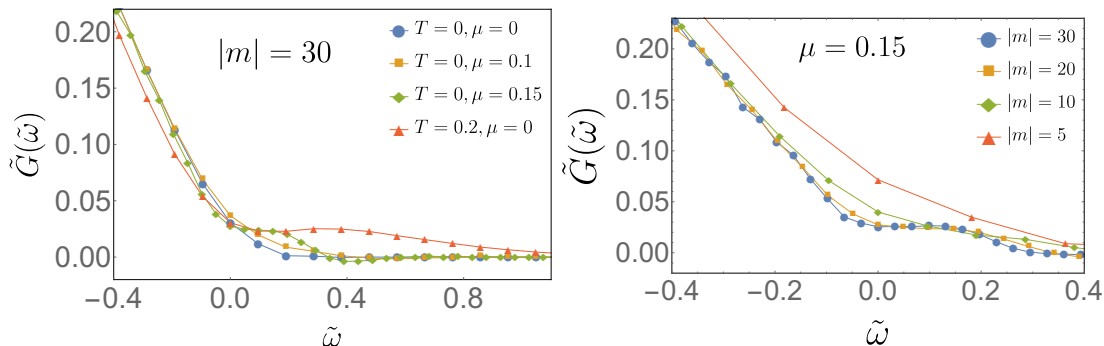

Figure 9: (color online) The influence of non-zero chemical potential and the non-zero temperate of atoms on the power spectrum of Wightman response function. Left panel: The distance to the horizon is fixed $|m| = 30$ which corresponds to the Unruh temperature $T_U \approx 0.015$ (see Fig.1). As expected, $\mu = 0.1$ and $\mu = 0.15$ data deviate from $T = 0$ curve only for small positive frequencies. Indeed, the $\mu = 0.1$ curve is very close to $T = 0$ result. The $\mu = 0.15$ curve exhibits more thermal behavior, still it less deviates form the perfect case then $T = 0.2$ curve. Right panel: The power spectrum of Wightman response function for a fixed nonzero chemical potential $\mu = 0.15$. Although the chemical potential is about an order of magnitude larger then the Unruh temperature, we can still distinguish the characteristic thermal properties of power spectra (see Eq.(60)).

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
